# Temporal assessment of N-cycle microbial functions in a tropical agricultural soil using gene co-occurrence networks

Marie Schaedel[1]*, Satoshi Ishii[2,3], Hao Wang[2], Rodney Venterea[2,4], Birthe Paul[5], Mupenzi Mutimura[6], Julie Grossman[1]

1 Department of Horticultural Science, University of Minnesota, St. Paul, MN, United States of America, 2 Department of Soil, Water, & Climate, University of Minnesota, St. Paul, MN, United States of America, 3 BioTechnology Institute, St Paul, MN, United States of America, 4 USDA-ARS, Soil & Water Management Research Unit, St. Paul, MN, United States of America, 5 Tropical Forages Program, International Center for Tropical Agriculture, Nairobi, Kenya, 6 Department of Animal Production, Rwanda Agriculture Board, Kigali, Rwanda

* schae659@umn.edu

## Abstract

Microbial nitrogen (N) cycling pathways are largely responsible for producing forms of N that are available for plant uptake or lost from the system as gas or leachate. The temporal dynamics of microbial N pathways in tropical agroecosystems are not well defined, even though they are critical to understanding the potential impact of soil conservation strategies. We aimed to 1) characterize temporal changes in functional gene associations across a seasonal gradient, 2) identify keystone genes that play a central role in connecting N cycle functions, and 3) detect gene co-occurrences that remained stable over time. Soil samples (n = 335) were collected from two replicated field trials in Rwanda between September 2020 and March 2021. We found high variability among N-cycle gene relationships and network properties that was driven more by sampling timepoint than by location. Two nitrification gene targets, hydroxylamine oxidoreductase and nitrite oxidoreductase, co-occurred across all timepoints, indicating that they may be ideal year-round targets to limit nitrification in rainfed agricultural soils. We also found that gene keystoneness varied across time, suggesting that management practices to enhance N-cycle functions such as the application of nitrification inhibitors could be adapted to seasonal conditions. Our results mark an important first step in employing gene networks to infer function in soil biogeochemical cycles, using a tropical seasonal gradient as a model system.

## Introduction

Soil nitrogen (N) cycling has critical implications for social, environmental, and production-oriented dimensions of agroecosystems. Asynchrony between crop demand and N supply can lead to N excesses and resulting deleterious environmental effects, such as nitrous oxide ($N_2O$) emissions and nitrate ($NO_3^-$) contamination of fresh water [1]. Soil microorganisms, including bacteria, archaea, and fungi, are responsible for transforming N into plant-available forms, as

**Data Availability Statement:** The datasets analyzed during the current study, as well as the code used to generate figures and conduct

analyses are publicly available on https://www.
github.com/schaedem/NiCE_Chip_Paper.

**Funding:** This work was undertaken as part of the
CGIAR Research Program (CRP) on Livestock. In
addition, it was supported by the OneCGIAR
Initiatives on Livestock, Climate and System
Resilience (LCSR). We thank all donors that
globally support our work through their
contributions to the CGIAR system. (Recipient:
International Center for Tropical Agriculture) This
work was also supported by the National Science
Foundation, Graduate Research Fellowship (no.
00074041) and the Doctoral Dissertation
Fellowship (University of Minnesota). (Ms. Marie
Schaedel).

**Competing interests:** The authors have declared
that no competing interests exist.

well as stabilizing N compounds into organic matter. Microbes also volatilize N compounds,
which results in significant agricultural soil N loss annually and accounts for up to 70% of
global $N_2O$ emissions [2]. A canonical understanding of the terrestrial N cycle consists of chro-
nological steps performed in sequence by soil microorganisms [3]. In the nitrification pathway,
N products starting with ammonium ($NH_4^+$) are sequentially oxidized, yielding nitrite ($NO_2^-$)
and/or nitrate ($NO_3^-$) as final products [4]. Denitrification is a complimentary pathway that
reduces N compounds, starting with $NO_3^-$ or $NO_2^-$ and ultimately yielding nitrous oxide
($N_2O$) or dinitrogen gas ($N_2$) as the final products (ref. 4; Fig 1).

This network of microbial N cycling processes does not always occur in an orderly and lin-
ear fashion. Copper-containing nitrite reductase (*nirK*) is involved in ammonia oxidation in
addition to its primary role in reducing nitrite ($NO_2^-$) to nitric oxide (NO) in denitrification
[5,6]. Nitrification can also be 'interrupted' when oxidative and reductive pathways co-occur
across small spatial and temporal scales [7]. Phylogenetically diverse microorganisms are
involved in soil N-cycling, with nitrification, denitrification, and N-fixation genes co-occur-
ring in many strains [8]. The reduction of an intermediate nitrification product ($NO_2^-$) to NO
and $N_2O$ can occur via nitrifier denitrification, which is performed by chemoautotrophic
ammonia oxidizers containing $NO_2^-$ reductase (*nxrB*) [9]. $N_2O$ reductase (*nosZ*) transcripts
have been found in non-denitrifying genera of aerobia bacteria such as *Gemmatimonas* [10].
This complexity, inherent in many microbial processes, obfuscates efforts to predict commu-
nity function from metagenomic or phylogenetic data.

Network analysis is increasingly used to analyze microbial populations and draw inferences
about complex community dynamics [11], including the degree to which nutrient cycling
genes co-occur in nature. Typically, networks are constructed with relative abundance or pres-
ence/absence data using a variety of techniques, including culture-based methods [12] and

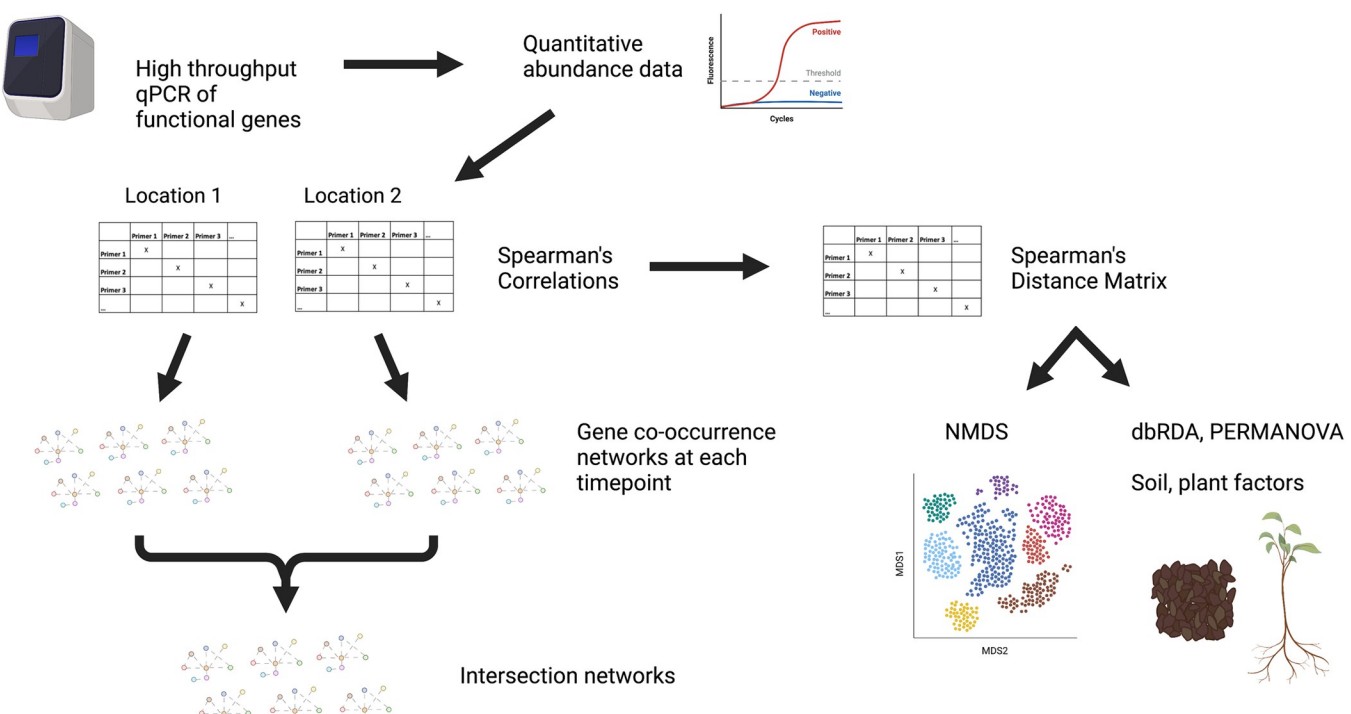

**Fig 1. Data collection and analysis workflow.**

**Table 1. Definition of network terms used in the present study.**

| Network Term | Definition/ Ecological Relevance | igraph function | References |
|---|---|---|---|
| Average path length | Average path length (L), is a measure of network efficiency defined as the average number of edges required to connect all possible pairs of nodes along the shortest path. | mean_distance | [16] |
| Co-occurrence | Co-occurrence is a predicted interaction between (in this paper) functional genes. We define co-occurrence using our cutoff value derived from a Spearman's correlation: rho > 0.75 and p<0.01. | | [17,18] |
| Node | In this paper, nodes represent unique primer pairs that were chosen to target functional genes and microbial groups involved in the N cycle. | | [19] |
| Edge | An implied relationship between nodes, which we obtained based on our criteria for co-occurrence. | get.edgelist | [19] |
| Density | Density is a measure of overall connectedness describing the complexity of a network. We calculated the ratio between the total edges summed across the entire network and the number of all possible edges. | edge_density | [20] |
| Clustering coefficient | On a scale of 0 to 1, clustering coefficient (also transitivity) describes the probability that the adjacent nodes are connected. It is an additional network-level statistic to describe complexity and connectedness. | transitivity | [21] |
| Betweenness Centrality | Betweenness is calculated as the number of shortest paths through a focal node. High betweenness centrality suggests a key role in connecting nodes across the network and is sometimes used to determine keystoneness. | betweenness | [18,22] |
| Closeness Centrality | Another measure of centrality, closeness centrality describes the proximity of a node to all other nodes in the network. It is calculated for each node by counting the number of steps required to reach every other node in the network. | closeness | [23,24] |
| Normalized Node Degree | A measure of connectivity, defined as the number of co-occurrences for each node. Node degree is calculated as the total number of edges per node normalized by the total number of nodes in the network. | degree | [18] |
| Modules | Groups of highly connected microbes. Networks with high modularity contain a greater number of modules. Genes within a module may have overlapping niches or functions. | edge. betweennness. community | [25,26] |
| Intersection Network | The intersection of two or more graphs; only links and nodes that are present in both graphs are retained in the consensus network. Consensus networks identify genes and co-occurrence interactions that are present across ecosystems, time, or space. | graph. intersection | [18] |

culture-independent methods such as metagenomic sequencing [13]. With abundance data in hand, network analysis can model positive and negative relationships, or co-occurrences, between microbial taxa [14]. Ecological inferences about microbial interactions and community functions can then be made. A positive co-occurrence between two nodes (Table 1) in a network could be due to niche overlap or commensalism, for instance when two organisms both respond to the presence of the same resource. Conversely, negative co-occurrences may be due to competitive exclusion or a predator-prey relationship [14]. Network analysis also aids in the identification of keystone taxa, defined as nodes that serve an important role in connecting different community functions [15].

Microbial network analyses commonly use whole-community sequencing methods (i.e., 16S rRNA gene or ITS2 amplicons) to ascribe functional attributes to specific taxa [27–29]. However, several unresolved challenges exist with this approach, including bias from compositional data [30]. In addition, assigning ecological functions to microbes based on taxonomic identification often leads to spurious conclusions [31]. Functional gene co-occurrence networks have not been widely used but could offer valuable insights into soil microbial functions such as nutrient cycling. Keystone genes, like keystone taxa, may be defined as those that have a large effect on community structure and function, for instance by connecting different enzymatic pathways [32]. Identifying keystone genes that link the nitrification and denitrification pathways will inform management decisions that enhance soil fertility and mitigate gaseous N losses.

High throughput qPCR (HT-qPCR) addresses the challenge of compositional data and functional inference in network analysis by directly quantifying gene targets with known functions [33,34]. Despite the extremely dynamic nature of microbial communities and networks [35], most studies to date have only collected data at a single point in time. The lower per-

sample cost of HT-qPCR compared to community sequencing methods can facilitate the study of dynamic networks that is crucial to informing soil management across a range of environmental, temporal, and geographic conditions [36]. In cases where the goal is to quantify microbial groups that perform a specific ecological function, HT-qPCR can be a cost-effective tool that is easier to interpret than amplicon sequence data. Recently, a HT-qPCR platform for N cycle gene was developed (27). The tool, called the N cycle gene evaluation (NiCE) chip, has been applied to assess gene abundances in forest and prairie soils [37,38]. The application of network analysis to HT-qPCR platforms is untested yet represents a promising low-cost opportunity to understand the functional potential of a microbial community.

In this study, we used the NiCE chip approach to probe the network structure of twenty-one N-cycle genes (S2 Table) across a seasonal gradient in a Rwandan cut-and-carry forage system. Tropical agroecosystems provide an excellent model to study dynamic gene networks because they are managed ecosystems with a high level of disturbance and are subject to distinct seasonal weather events. Perturbations in soil conditions such as changes in moisture and temperature vary dramatically in rainfed tropical farming systems [39] with strong effects on microbial community composition and functional potential [40]. For instance, the duration and intensity of a drought period (such as a typical tropical dry season) affects microbial resource use efficiency and function, with implications for potential N loss [41,42]. In addition, complete aboveground defoliation is characteristic of cut-and-carry forage systems in densely populated East Africa [43] and induces belowground shifts in C allocation that impact soil microbial abundance and activity [44,45].

Our overall goal was to demonstrate the utility of functional gene networks in studying dynamic soil biogeochemical processes and identifying temporally important N-cycle gene interactions in a managed forage system in Rwanda. We chose sampling timepoints that captured two different plant growth stages (anthesis and early post-harvest regrowth) across a six-month period that included both the rainy and dry seasons.

Given that microbial activity is affected by both edaphic and plant factors, we expected to see significant differences in gene-gene associations and network structure between timepoints. Specifically, we aimed to 1) characterize temporal trends in N-cycle functional gene interactions, 2) identify keystone genes, and 3) detect functional gene co-occurrences that remained consistent across all sampling timepoints. We hypothesized that 1) higher soil moisture in the rainy season would stimulate greater network connectivity compared to the dry season, 2) keystone genes would vary seasonally, and 3) nitrification and denitrification pathways would remain linked by one or more genes irrespective of season, location, or plant growth stage.

## Methods

### Site description

This study was conducted at two locations in Rwanda (Karama and Rubona), each with a randomized block design consisting of four replicates. Plant treatments included perennial forage grasses and annual maize grown with or without a perennial legume intercrop. Karama (1˚31'28.5"S 30˚21'52.9"E) is in the low altitude (1400 m) Eastern Province of Rwanda. Karama is a tropical savanna with a mean annual rainfall of 904 mm and a mean annual temperature of 21.0˚C. Rubona (2˚28'59.9"S 29˚46'31.9"E) is located in the Huye Province, which is a mid-altitude (1700 m) tropical savanna. The mean annual temperature in Nyanza is 20.0˚C and the mean annual rainfall is 1077 mm. Rainfall during the sampling period and soil properties for each site are provided in S2 Fig.

## Soil sampling

Soils samples were collected immediately (1–2 days) prior to forage harvest and two weeks after harvest for a total of three harvests (6 sampling timepoints) between September 2020 and March 2021. Forages were harvested and total aboveground biomass removed about every 8 weeks, just before flowering. Soil cores were collected to a depth of 12 cm using sterile plastic tubes inserted vertically into the ground, using a new set of tubes between plots to limit microbial cross-contamination. This method of sterile sampling was chosen because ethanol was not easily available in Rwanda to clean a soil probe during sampling at the research sites. Soil samples were homogenized in a sterile plastic bag and passed through a 2mm sieve.

## Soil assays

Soil samples were analyzed at the University of Minnesota (UMN, St Paul, MN, USA) within two weeks of the sample collection date. The soils were shipped in insulated containers with ice packs and were still cold upon their arrival to UMN. At each sampling period, soils were analyzed for permanganate oxidizable carbon (POXC) [46], mineral nitrogen ($NH_4^+$-N, $NO_3^-$N) [47], and potentially mineralizable nitrogen (PMN) [48]. Nitrification potential (NP) [49] and denitrification enzyme activity (DEA) [50] were performed using modified versions of previously described methods. DEA was log-transformed to meet normality assumptions of subsequent analyses. Soil pH was measured from slurries prepared by mixing 10g air dried soil and 30ml deionized $H_2O$. Gravimetric water content (GWC) was determined by obtaining the mass difference between field-moist and air-dried soils.

## DNA extraction

Upon arrival at the UMN, a portion of each soil sample was repackaged and stored at -20˚C for eventual DNA extraction. Extractions were performed on 0.25 g soil using the DNeasy PowerSoil DNA Kit (Qiagen, Hilden, Germany) according to the manufacturer's protocol. The final elution step was modified such that 50 ul of elution buffer was used to obtain higher DNA concentrations.

## Conventional 16S rRNA gene qPCR

Conventional quantitative polymerase chain reaction (qPCR) of the bacterial 16S rRNA gene was used to assess DNA extraction quality and test for differences in bacterial abundance among treatments, collection timepoint, and location [51]. We used a previously validated protocol for SYBR green qPCR [52]. A full description of these methods can be found in S1 Text.

## NiCE chip

We used the SmartChip platform (Takara Bio, Shiga, Japan) to comprehensively assess N-cycle gene dynamics in Rwandan pasture soils using the NiCE chip approach described in ref. (26). The SmartChip system can perform 5,184 qPCR reactions simultaneously each in a 100-nl reaction well. Assays for the NiCE chip were chosen to cover major microbial N transforming reactions (nitrification, denitrification, N fixation, anammox, and DNRA) [38,53]. We ran 38 qPCR assays of N-cycle gene targets for 128 samples on each chip simultaneously. N-cycle gene targets in the soil samples were calculated from their given cycle threshold (Ct) values and the assay standard curves using linear regression as described previously [54]. Twenty-one gene targets were retained following our data cleaning procedure. For all subsequent analyses, we used gene abundance data normalized to log copes g$^{-1}$ soil. A complete list of primers and detailed methods can be found in S2 Table.

## Network analysis

Network analysis was conducted in R (version 4.0.3) using the igraph package [55]. We separated samples by location and timepoint for our initial analysis. We used Spearman's correlations of gene abundances normalized to copies $g^{-1}$ soil to identify associations among NiCE chip assays. Spearman's correlation is more successful than Pearson's in detecting monotonic, non-linear relationships because it does not have a normality requirement [18].

We created undirected co-occurrence networks based on the strength ($\rho$) and significance of gene correlations, using stringent cut-off values of $|\rho| > 0.75$ and $P < 0.01$ based on similar values used in recent studies [18,56]. We created twelve co-occurrence networks, one for each timepoint at each location (Fig 1). We did not identify any negative correlations at the level of $\rho \leq -0.75$, so our networks consisted only of positive associations. To correct for false discovery rate from multiple pairwise correlation tests, we used the Benjamini and Hochberg p-value correction [57] using adjust.corr in the bcdstats R package [58].

To focus on temporal trends and identify shared gene co-occurrences between locations, we collapsed these twelve networks into six networks, each of which represents the intersection by location. These intersection networks retained only nodes and edges that were present in both original networks. Lastly, we simplified the networks to remove loops (nodes with zero edges that are correlated to themselves). Network statistics (Table 1) for the intersection networks were generated using the igraph package.

## Statistical analysis

Spearman's distance was calculated using spearman.dist from the BioDist package, as described in ref. (45). We used PERMANOVA tests to investigate the effect of season and location on N-cycle co-occurrence structure with adonis from the vegan package [59]. The null hypothesis for PERMANOVA using Spearman dissimilarity matrices is that the centroids of each group are the same. PERMANOVA results were visualized using NMDS ordinations generated in vegan with monoMDS. Finally, distance-based redundancy analysis (dbRDA) was employed to quantify the contribution of soil physiochemical factors to gene co-occurrence patterns using the dbrda function from vegan. Finally, Spearman's correlations were used to identify relationships between enzyme activity, soil properties, and functional gene abundance.

# Results

## Drivers of N-cycle gene co-occurrence

Most N-cycle genes, aside from archaeal *amoA* (measured by Arch-amoA_F assay), had strong position correlations with each other across sampling periods and locations. In general, N-cycle genes were weakly correlated with the abundance of 16S rRNA gene ($0 < \rho < 0.34$; S4 Fig). Nitrification potential (NP) was negatively correlated with all NiCE chip assays ($-0.31 < \rho < -0.145$) except for Arch-amoA_F ($\rho = 0.24$; S3 Fig). Likewise, denitrification enzyme activity (DEA) was positively correlated with Arch-amoA_F ($\rho = 0.29$, $P < 0.05$) but negatively correlated with all other assays ($-0.46 < \rho < -0.11$).

We next analyzed the effects of location and sampling date on gene co-occurrence (Fig 1). Timepoint, location, and the timepoint x location interaction were highly significant in accounting for differences in gene co-occurrence patterns (PERMANOVA, $P < 0.001$ for all), although timepoint explained the most variation (timepoint$_{R2} = 0.40$, location$_{R2} = 0.07$, timepoint x location$_{R2} = 0.04$). Co-occurrence patterns in rainy season samples were generally more similar to each other than to dry season samples (Fig 2). The first two timepoints,

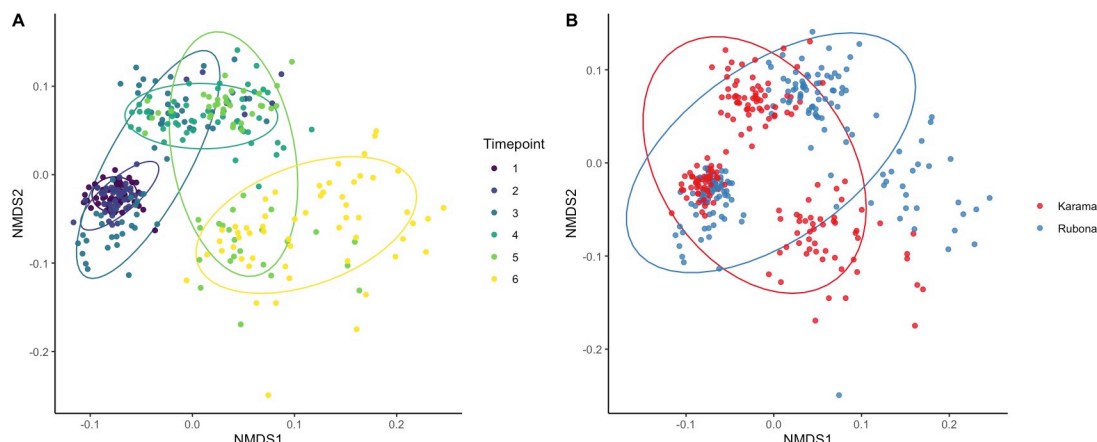

**Fig 2. NMDS ordinations of gene co-occurrences at two locations in Rwanda across 6 sampling periods using a Spearman's dissimilarity matrix.** Gene co-occurrences were strongly driven by seasonal factors (A) and secondarily by location (B).

corresponding to the late dry season, did not differ significantly in co-occurrence patterns ($P > 0.05$). Timepoints two and three, corresponding to the late dry season and early rainy season, respectively, also did not differ in terms of gene co-occurrence ($P > 0.05$). All other pairwise comparisons between timepoints were significant at $P<0.01$.

We next identified globally important soil physiochemical properties that accounted for the structure of gene-gene interactions across time. Without accounting for the main factors of sampling timepoint or location, the most parsimonious dbRDA model identified with forward selection found that gene co-occurrence patterns were significantly associated with $NO_3^-$-N, $NH_4^+$-N, PMN, GWC, POX-C, pH, and DEA (Fig 3). This full model accounted for 38.2% of observed variation in gene-gene associations ($P<0.001$). The first axis, which accounted for 29.4% of variation, related to dispersion according to $NO_3^-$-N, GWC, and DEA. PMN, representing readily available pools of organic N, varied mainly along the second axis (7.4% of variation). While pH, POXC and $NH_4^+$-N were retained in the model, they appeared to play a weaker explanatory role in accounting for gene-gene associations based on the vector lengths. We also ran dbRDA within each timepoint (S6 Fig) to understand how the influence of soil factors on gene co-occurrence patterns shifted over time. While rainy season gene co-occurrences were strongly influenced by soil properties, parsimonious dbRDA models did not retain any soil variables within the dry season timepoints.

## Temporal trends in co-occurrence network structure

Comparison of network structure across time indicates that the functional N-cycling community was highly dynamic, with network properties changing over the sampling periods corresponding to season and plant growth stage (Fig 4). N-cycle gene network density and clustering coefficient (Table 1) decreased overall between the dry and the rainy season. We observed the greatest network density in the first two sampling timepoints, which occurred in the dry season (Table 2). The densest network (timepoint 1) and the sparsest network (timepoint 6) consisted of a single module that contained all the nodes in the network. Regardless of season, N-cycling gene networks tended to be denser at anthesis ($0.52 \pm 0.17$) than during early post-harvest regrowth ($0.36 \pm 0.04$), although this was not significant with our sample size.

We observed fewer co-occurrences involving fungal and archael genes in rainy season networks compared to the late dry season (Fig 4). Except for the final rainy season timepoint,

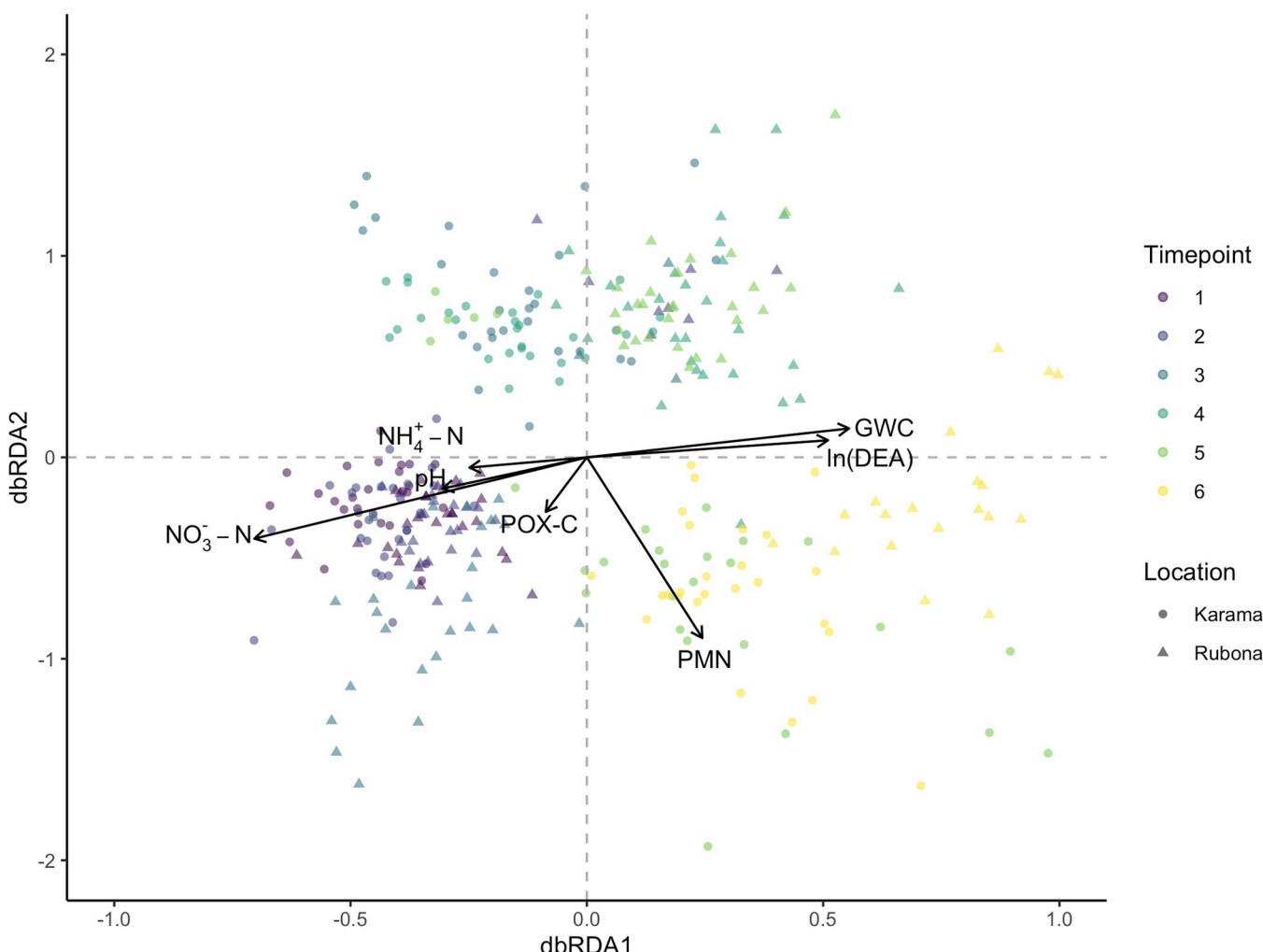

**Fig 3. Distance-based redundancy analysis of gene co-occurrences constrained by soil properties (0–12 cm) at two locations in Rwanda across 6 sampling periods.** A Spearman's dissimilarity matrix was used as the response factor. A forward selection procedure identified the most parsimonious set of explanatory variables to retain in the model. Vectors represent soil properties retained in the model, with vector length corresponding to explanatory power. Axes 1 and 2 explained 29.4% and 7.4% of the observed variation, respectively. The full model explained 38.2% of observed variation in gene co-occurrences (P<0.001).

AOA had fewer degrees than AOB across time, with beta-proteobacterial *amoA* (Beta_amoA assay) consistently having a higher degree. Similarly, fungal *nirK* had a similar degree as bacterial *nirK* genes (nirK_876 and nirK_FlaCu assays) in the first dry season timepoint, but in subsequent timepoints had only a single degree or was removed from the network entirely. By the late rainy season (timepoints 5 and 6), only functional genes from the nitrification and denitrification pathways were retained in our intersection networks. Nitrogen fixation gene (*nifH* measured by the nifH_IGK3 assay) and comammox (comaB assay) did not appear in rainy season networks past timepoints 3 and 2, respectively. The genes for dissimilatory nitrate reduction (*napA*, *narG*) and nitrite reduction to ammonia (*nrfA*) did not co-occur with other N-cycle genes at any timepoint.

Finally, we compared features of our intersection gene networks with Erdös-Rényi random networks containing the same number of nodes and edges. The properties we observed in our gene networks (Table 2) were more similar to food webs and pollinator-plant networks [21,60] than to a previously reported functional gene network [61], most likely due to the small

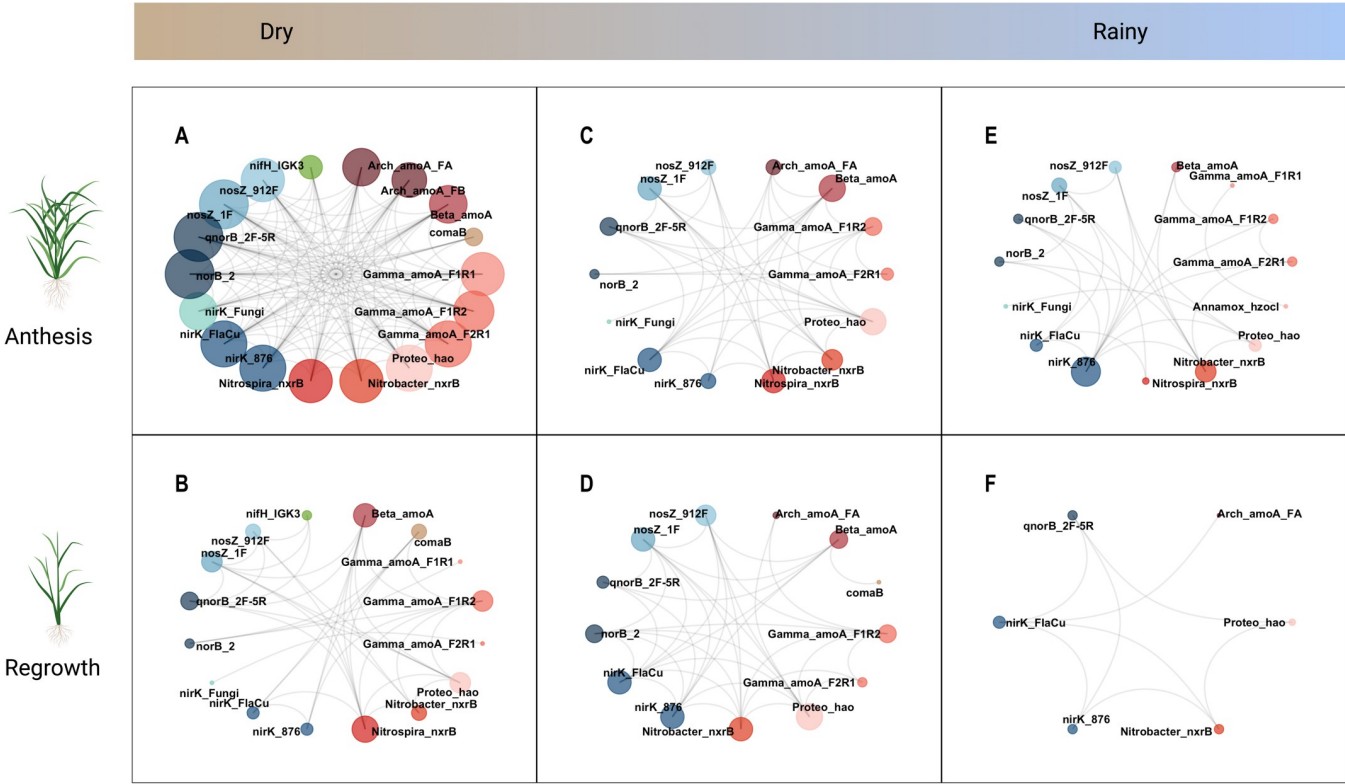

**Fig 4. Non-directed intersection networks of N-cycle functional gene co-occurrences defined as a Spearman correlation of $|\rho| > 0.75$ and P < 0.01 (see methods).** These networks display seasonally driven co-occurrences that were common to both sampling locations. Sampling timepoints occurred in the late dry season (A, B), mid rainy season (C, D), and late rainy season (E, F). Networks in the top row (A, C, E) correspond to plant growth at the anthesis stage while networks in the bottom row (B, D, E) represent early-stage regrowth at 2 weeks post-harvest. Nodes representing nitrification gene targets are shown in warm colors, while denitrification gene targets are shown in cool colors. Redundant targets for the same gene and organism were given the same color.

network sizes. While our first five intersection networks share similarities with small-world networks [62], the final rainy season network does not differ significantly from random network properties (Table 2). It is likely difficult to extract meaningful structural features from this final network because of its sparsity and size (8 nodes).

## Keystone N-cycle genes

We looked for keystone genes that played a role in connecting network functions to identify putative keystones at each timepoint. Betweenness centrality, closeness centrality, and

**Table 2. Properties of observed intersection networks and Erdös-Renyi random networks of the same size.**

| Timepoint | Season | Density | $C_{obs}$ | $C_{random}$ | $L_{obs}$ | $L_{random}$ | RR* | LRR** |
|---|---|---|---|---|---|---|---|---|
| 1 | Late dry | 0.83 | 0.89 | 0.73 ± 0.01 | 1.17 | 1.26 ± 0 | 1.22 | 0.2 |
| 2 | Late dry | 0.28 | 0.62 | 0.30 ± 0.05 | 1.86 | 1.83 ± 0.04 | 2.07 | 0.73 |
| 3 | Mid rainy | 0.46 | 0.73 | 0.44 ± 0.04 | 1.85 | 1.56 ± 0.02 | 1.66 | 0.51 |
| 4 | Mid rainy | 0.42 | 0.68 | 0.47 ± 0.04 | 1.68 | 1.53 ± 0.02 | 1.45 | 0.37 |
| 5 | Late rainy | 0.25 | 0.46 | 0.23 ± 0.06 | 1.86 | 2.03 ± 0.09 | 2 | 0.69 |
| 6 | Late rainy | 0.38 | 0.38 | 0.42 ± 0.15 | 1.53 | 1.52 ± 0.06 | 0.9 | -0.11 |

C = clustering coefficient, L = average path length, RR (response ratio) = $C_{obs}/C_{random}$, LRR (log response ration) = $\ln(C_{obs}/C_{random})$.

normalized node degree are measures that can indicate the relative importance of a node in a network (Table 1). To identify putative keystone N-cycle genes, we selected assays that had the greatest value in at least two of these measures in the nitrification and denitrification pathways at each timepoint (S4 Table). We decided to focus only on nitrification and denitrification because genes outside of these pathways, such as *nifH*, generally did not have central positions in the networks.

Nitrification genes on average had greater betweenness centrality (4.45 ± 0.14) than denitrification genes (3.12 ± 0.15). Unlike keystone nitrification genes whose network centrality tended to remain consistent over multiple timepoints, denitrification gene centrality was highly transient. For instance, among nitrification genes, *amoA* of *Gammaproteobacteria* (Gamma-amoA_F2R1 assay) had the highest betweenness centrality across both the second and third timepoints and *hao* (Proteo_hao assay) had the highest betweenness for both the fourth and fifth timepoints. In contrast, there were no denitrification genes that retained a central network position for more than one timepoint (Fig 5).

## Stable functional gene co-occurrence relationships

To identify seasonally stable N-cycle gene co-occurrence relationships that were present across all timepoints and both locations, we found the intersection of the previously described set of six networks. This resulted in the creation of a single network that included significant gene co-occurrences that occurred in all locations and timepoints. The final intersection network consisted of two unconnected diads (one each for nitrification and denitrification assays) (S7 Fig). Because our NiCE chip protocol contained several sets of redundant assays that were

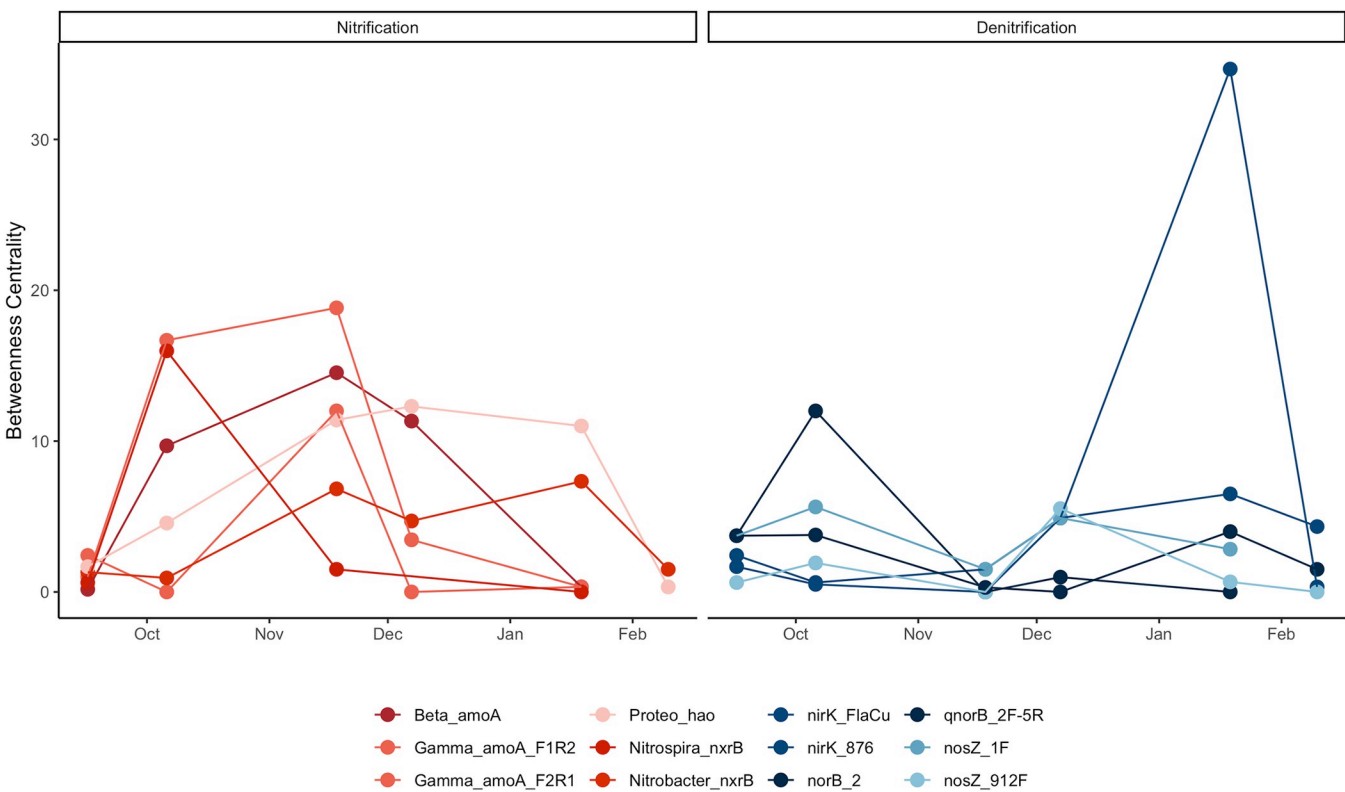

**Fig 5. Betweenness centrality of putative keystone genes varies by time in tropical rainfed pasture soils.** We selected a subset of nitrification and denitrification gene targets, shown here, and identified them as keystones based on node degree, betweenness centrality, and closeness centrality.

designed to target the same gene and organism, we expected these redundant primer sets to co-occur in all networks. However, only one pair of redundant assays (nirK_FlaCu and nirK_876) was present in the final consensus network, indicating that these two qPCR assays yielded similar results across the duration of the experiment. The second diad in the final network contained Nitrobacter_nxrB and Proteo_hao. Contrary to our expectations, *amoA* did not retain a stable co-occurrence with any other gene, nor did we not identify a stable link between the nitrification and denitrification pathways.

## Discussion

### Understanding drivers of N-cycle gene co-occurrence

Our temporal analysis of gene networks confirmed known properties of N-cycling microorganisms and generated novel hypotheses about their ecology in an East African cut-and-carry perennial forage system. While it is difficult to attribute network features directly to plant growth stage, the differences we observed in network structure corresponded to seasonal-temporal trends in soil moisture and mineral N availability. Therefore, soil and environmental factors likely played a major role in shaping microbial N-cycle gene co-occurrences.

Most variation in N-cycle gene associations on the first dbRDA axis was driven by soil characteristics relating to substrate availability ($NH_4^+$-N, $NO_3^-$N) and factors (GWC, pH) that are known to be globally important drivers of microbial metabolism (Fig 3). Gene co-occurrences in the late dry season varied most dramatically from co-occurrences in the late rainy season, with the main differences evident in the availability of mineral N sources, water content, and DEA. Associations among genes from mid rainy season timepoints fell in-between. Soil pH was a significant term in our model because it accounted for site-specific differences: the average pH in Rubona was almost a full unit lower than Karama (S1 Table). Nonetheless, pH played a relatively small role compared to other dynamic soil properties that changed during the seasonal transition. Strikingly, none of the soil properties we measured accounted for variation in gene co-occurrence structure in the dry season (S6 Fig). In line with previous research on microbial responses to drought stress, this result most likely reflects a state of metabolic dormancy requiring moisture-induced resuscitation over time [63]. The strong correlation we observed across all N-cycle gene targets is supported by previous work that found a phylogenetic basis for resuscitation strategies following wet-up of dry soil [64].

The second dbRDA axis corresponded to pools of organic substrates (POXC, PMN) that are easily metabolized by the soil microbial community. Both POXC and PMN are dynamic soil properties [46,65], and could therefore account for root-induced changes to gene abundance. Due to multiple confounding variables associated with time, we could not directly test the contribution of plant growth stage to N-cycle gene interactions. However, we propose that POXC and PMN, which did not demonstrate clear seasonal trends as with other variables (GWC, $NH_4^+$-N, $NO_3^-$N) may account for root-derived inputs to the system. While research on plant-soil feedbacks in tropical perennial forages is limited, a related study on the perennial grain crop Kernza (*Thinopyrum intermedium*) found that harvested treatments experienced increases of 73% and 49% in root biomass and POXC compared to unharvested controls, respectively [66]. Dynamic belowground responses to biomass removal in our rainfed cut-and-carry system aligned with our expectation that root-derived carbon may directly and indirectly affect the function of the N-cycle community.

### N-cycle co-occurrence networks are seasonally dynamic

The strong positive correlations we identified among the NiCE chip assays suggest a high degree of niche overlap among members of the N cycling community. This is true especially in

the dry season, when we observed the greatest number of co-occurrences as well as the greatest network density.

AOB tended to co-occur more frequently with denitrification genes than AOA across time. This could mean either that bacterial N-cyclers share a similar niche space (e.g. linked nitrification and denitrification) or that an organism containing the genetic capability to perform both nitrification and denitrification (e.g. *Nitrosomonas europoea*) is present in high numbers [8,56,67]. Dry soil conditions increase the rate of cell-to-cell microbial interactions, which could explain the higher number of co-occurrences in the dry season [68]. Additional sequencing or qPCR work is needed to clarify the spatiotemporal dispersion of the N-cycling community across time in this system.

Network sparsity corresponded with higher enzymatic potential in the rainy season. This result contradicted our expectation that higher soil moisture in the rainy season would increase network complexity by increasing total microbial abundance. In the dry season, we observed higher levels of mineral N (S8 Fig) may have been unavailable for microbial activity due to limitations in diffusion and transport [59]. While the rainy season stimulated microbial enzymatic potential, we observed significant declines in mineral N due to putative topsoil leaching, erosion, and microbial uptake. Thus, a different set of constraints in the rainy season may have selected for more competitive taxa, resulting in smaller networks.

Network analysis could be useful in sustainable nutrient management to identify the time at which management interventions may have the greatest impact, for instance when gene networks display the greatest density. Given that highly nested and complex networks are unstable [69], applying nitrification inhibitors in the dry season (higher connectivity, lower activity) vs. the rainy season (lower connectivity, higher activity) may effectively prevent N losses from denitrification during the seasonal transition. We did not find any stable relationships between the nitrification and denitrification pathways. This suggests that inhibition of one or more nitrification genes may not reliably influence denitrification processes across all soil conditions.

### Leveraging keystone genes for adaptive nutrient management

Our findings suggest that keystone functions of the N-cycling microbial community were dynamic in response to temporal and environmental stimuli, rather than static over the seasonal transition. Overall, nitrification genes had higher average betweenness in our intersection networks. This result was expected, as nitrification genes occur upstream of denitrification genes in the N-cycle and result in products such as NO and $NO_2$ that can directly enter the denitrification pathway [8].

The network positions of AOA were markedly different from AOB throughout our sampling period. Notably, AOA have a capacity for heterotrophic growth and are not obligately tied to nitrification as an energy source, as with AOB [70]. This observation concurs with previous research that has found AOB to be responsive to environmental perturbations. For instance, AOB functional gene abundance responds rapidly to N availability [21] and changes in moisture and temperature [71,72]. The keystoneness of AOB in response to seasonal changes has not been previously reported but concurs with our understanding of how this group of nitrifiers responds to environmental stimuli. Our results imply that interventions targeting AOB, rather than AOA, are more likely to influence downstream denitrification processes that generate $N_2O$, NO, and $N_2$.

Denitrification genes were more responsive to environmental and temporal factors than nitrification targets. This agrees with previous research that has found denitrification genes, specifically *nirK*, to be sensitive to changes in soil water content [73]. Denitrifiers are subject to a complex set of interactions, including plant community, soil chemistry, and interactions

with other microbial communities that are not fully understood [74–76]. While we expected denitrification gene network centrality to increase in the rainy season, our results suggest that this keystone position fluctuated dramatically and was not sustained even under high moisture conditions.

## Limitations

We used genomic DNA copy number to infer function of the N-cycling microbial community, which does not correlate to transcription of our target genes *in situ*. Nevertheless, HT-qPCR measurement of functional genes is a more direct way to infer community function than through other commonly used practices such as functional predictions based on the 16S rRNA gene amplicon sequencing. HT-qPCR with redundant assays targeting the same gene or group of organisms is especially useful in novel systems with limited prior knowledge about microbial community composition. The redundant assays in this study demonstrated markedly different abundance and co-occurrence patterns, underscoring the often-arbitrary nature of primer selection employed in similar studies [72,73,77] that can dramatically influence conclusions.

## Conclusion

Our work identified keystone N-cycle genes and co-occurrence relationships that remained stable across a dry to rainy season transition and two different plant growth stages. Temporal shifts in network structure suggest that while higher soil moisture in the rainy season may increase the metabolic activity of the N-cycling community, other factors such as competition or resource scarcity may destabilize gene-gene relationships and thus total potential N-cycle pathways. Controlled growth chamber studies are needed to test these proposed mechanisms. We have shown, however, that network analysis of functional genes is useful in generating hypotheses pertaining to microbial community function and informing adaptive soil nutrient management.

## Supporting information

**S1 Checklist. Inclusivity in global research statement.**
(PDF)

**S1 Fig. Map of the two study locations (Karama and Rubona) in Rwanda.** This map was generated using the 'maps' package in R, which imports data from the public domain Natural Earth project.
(TIFF)

**S2 Fig. Rainfall during the sample collection period.**
(TIFF)

**S3 Fig. Correlogram of gene abundances.** An 'X' denotes a non-significant Spearman's correlation at the level of $p = 0.05$. Made with the 'ggstatsplot' package in R.
(TIFF)

**S4 Fig. Spearman's correlations between gene targets and soil features.**
(TIFF)

**S5 Fig. Heatmap of gene target abundances in Karama and Rubona.** Gene abundances were normalized to $\log_{10}$ copies per gram of soil.
(TIFF)

**S6 Fig. Distance-based redundancy analysis within sampling timepoint.** A Spearman's dissimilarity matrix was used as the input for each dbRDA.
(TIFF)

**S7 Fig. Intersection network for all locations and timepoints.** This networks displays N cycle gene co-occurrences that were common across all sampling timepoints and both locations. Nodes representing nitrification gene targets are shown in warm colors, while denitrification gene targets are shown in cool colors. Redundant targets for the same gene and organism were given the same color.
(TIFF)

**S8 Fig. Soil mineral nitrate in Karama and Rubona during the sampling period.**
(TIFF)

**S1 Table. Soil characteristics of the two study locations.** Soils were sampled at the beginning of the sampling period (September, 2020) for routine analyses.
(XLSX)

**S2 Table. List of primers used in NiCE Chip HT-qPCR.**
(DOCX)

**S3 Table. Functional gene network properties at each location x timepoint.**
(XLSX)

**S4 Table. Putative keystone nitrification and denitrification gene targets by timepoint.**
(XLSX)

**S1 Text. Detailed methods.**
(DOCX)

## Acknowledgments

We would like to thank the Rwandan Agricultural Board and Paulin Mutanguha for their support in maintaining field trials. We would also like to thank Solomon Mwendia and An Notenbaert from the International Center for Tropical Agriculture (CIAT) for providing guidance and logistical support. Bonsa Mohamed and Scott Mitchell at the University of Minnesota helped with sample processing and provided technical support.

## Author Contributions

**Conceptualization:** Marie Schaedel, Satoshi Ishii, Birthe Paul, Julie Grossman.

**Data curation:** Marie Schaedel.

**Formal analysis:** Marie Schaedel, Satoshi Ishii.

**Funding acquisition:** Birthe Paul, Mupenzi Mutimura.

**Investigation:** Marie Schaedel.

**Methodology:** Satoshi Ishii, Hao Wang, Rodney Venterea, Mupenzi Mutimura, Julie Grossman.

**Project administration:** Birthe Paul, Mupenzi Mutimura.

**Supervision:** Julie Grossman.

**Validation:** Hao Wang, Rodney Venterea.

**Writing – original draft:** Marie Schaedel.

**Writing – review & editing:** Satoshi Ishii, Hao Wang, Rodney Venterea, Birthe Paul, Mupenzi Mutimura, Julie Grossman.

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
