## [Decision Letter · Decision Letter 0]

21 Oct 2022

PONE-D-22-22455Temporal assessment of N-cycle microbial functions in a tropical agricultural soil using gene co-occurrence networksPLOS ONE

Dear Dr. Schaedel,

Thank you for submitting your manuscript to PLOS ONE. After careful consideration, we feel that it has merit but does not fully meet PLOS ONE’s publication criteria as it currently stands. Therefore, we invite you to submit a revised version of the manuscript that addresses the points raised during the review process.

We look forward to receiving your revised manuscript.

Kind regards,

Upendra Kumar, Ph.D.

Academic Editor

PLOS ONE

Journal Requirements:

"The authors declare no competing interests."

9. We note that Figure S1 in your submission contain map image which may be copyrighted. All PLOS content is published under the Creative Commons Attribution License (CC BY 4.0), which means that the manuscript, images, and Supporting Information files will be freely available online, and any third party is permitted to access, download, copy, distribute, and use these materials in any way, even commercially, with proper attribution. For these reasons, we cannot publish previously copyrighted maps or satellite images created using proprietary data, such as Google software (Google Maps, Street View, and Earth). For more information, see our copyright guidelines: http://journals.plos.org/plosone/s/licenses-and-copyright.

a. You may seek permission from the original copyright holder of Figure S1 to publish the content specifically under the CC BY 4.0 license.  

Reviewers' comments:

Reviewer's Responses to Questions

**Comments to the Author**

1. Is the manuscript technically sound, and do the data support the conclusions?

Reviewer #1: Yes

Reviewer #2: Yes

2. Has the statistical analysis been performed appropriately and rigorously? 

Reviewer #1: Yes

Reviewer #2: Yes

3. Have the authors made all data underlying the findings in their manuscript fully available?

Reviewer #1: Yes

Reviewer #2: Yes

4. Is the manuscript presented in an intelligible fashion and written in standard English?

Reviewer #1: Yes

Reviewer #2: Yes

5. Review Comments to the Author

Reviewer #1: Full treatment details need to be given like no. of treatments, names of forage grasses, legumes.

Results section needs to be elaborated by mentioning keystone genes identified for nitrogen cycle processes.

Figures are not clearly visible, hence prepare high resolution figures for clear visibility.

Table 3 is missing.

At some places typo errors are there like copes for copies, thoroughly check the entire manuscript.

Supplemental table 3 and 4 are not cited in the text.

Reference style is not uniform, hence maintain uniformity.

In requirement of the above, the manuscript needs major revision.

Reviewer #2: This manuscript aims to elucidate and characterize temporal changes in nitrogen cycle functional genes, which play a vital role in connecting N cycle functions in agricultural fields, using new technique of HT-qPCR, which is a promising low-cost tool to understand the functional potential of a microbial community. The results are convincing, experiments were properly designed and the findings have important implications towards the change in microbial gene functions related to N-cycle, which is most responsible for availability of nitrogen for crop growth. As the nitrogen is the most crucial element for crops especially in tropical agroecosystems, which is highly managed manmade ecosystem.

Overall, the manuscript is fit for publication. Result portions are written well except at some places written in such that it is difficult to understand the outcome. Result should more clearly written. Conclusions should be written in more elaborate manner. In conclusions, mention in detail about novelty of this work compared to what has already been published to emphasize on the impact of this study.

6. PLOS authors have the option to publish the peer review history of their article (what does this mean?). If published, this will include your full peer review and any attached files.

Reviewer #1: No

Reviewer #2: **Yes: **Himani priya

---

## [Author Response · Author response to Decision Letter 0]

6 Jan 2023

Thank you for the feedback raised in the review process. We respond to specific comments in depth in the attached 'Response to Reviewers' file.

---

## [Editor Report · Decision Letter 1]

24 Jan 2023

Temporal assessment of N-cycle microbial functions in a tropical agricultural soil using gene co-occurrence networks

PONE-D-22-22455R1

Dear Dr. Schaedel,

We’re pleased to inform you that your manuscript has been judged scientifically suitable for publication and will be formally accepted for publication once it meets all outstanding technical requirements.

Kind regards,

Upendra Kumar, Ph.D.

Academic Editor

PLOS ONE
---

## [Editor Report · Acceptance letter]

5 Feb 2023

PONE-D-22-22455R1 

Temporal assessment of N-cycle microbial functions in a tropical agricultural soil using gene co-occurrence networks 

Dear Dr. Schaedel:

I'm pleased to inform you that your manuscript has been deemed suitable for publication in PLOS ONE. Congratulations! Your manuscript is now with our production department. 

Kind regards, 

on behalf of

Dr. Upendra Kumar 

Academic Editor

PLOS ONE